# Bioavailability of Celecoxib Formulated with Mesoporous Magnesium Carbonate—An In Vivo Evaluation

**DOI:** 10.3390/molecules27196188

**Published:** 2022-09-21

**Authors:** Teresa Zardán Gómez de la Torre, Tuulikki Lindmark, Ocean Cheung, Christel Bergström, Maria Strømme

**Affiliations:** 1Department of Materials Science and Engineering, Uppsala University, Box 35, 751 03 Uppsala, Sweden; 2Disruptive Pharma AB, Dag Hammarskjölds Väg 54B, 751 83 Uppsala, Sweden; 3Department of Pharmacy, Drug Delivery, Uppsala University, Box 580, 751 23 Uppsala, Sweden

**Keywords:** mesoporous materials, magnesium carbonate, poorly soluble drug, celecoxib, drug release, bioavailability

## Abstract

An attractive approach to increase the aqueous apparent solubility of poorly soluble drugs is to formulate them in their amorphous state. In the present study, celecoxib, a poorly soluble drug, was successfully loaded into mesoporous magnesium carbonate (MMC) in its amorphous state via a solvent evaporation method. Crystallization of celecoxib was suppressed, and no reaction with the carrier was detected. The MMC formulation was evaluated in vitro and in vivo in terms of oral bioavailability. Celebra^®^, a commercially available formulation, was used as a reference. The two celecoxib formulations were orally administrated in male rats (average of *n* = 6 animals per group), and blood samples for plasma were taken from all animals at different time points after administration. There was no statistical difference (*p* > 0.05) in AUC_inf_ between the two formulations. The results showed that MMC may be a promising drug delivery excipient for increasing the bioavailability of compounds with solubility-limited absorption.

## 1. Introduction

Poor aqueous solubility of small molecular drug candidates constitutes an increasing challenge in drug development. About 40% of currently marketed drugs and up to 75% of compounds under development suffer from low aqueous solubility [1,2,3]. This may result in poor bioavailability and limited therapeutic effect if administered orally, which is the preferred route of administration. To circumvent this problem, different formulation strategies are used in drug products, including the reduction in drug particle size, salt forms of drugs, and the use of surfactants and lipid-based formulations [1,4,5]. These strategies have their respective benefits and limitations, and since different active pharmaceutical ingredients (APIs) have different physiochemical properties, there is no general solution to increase the solubility of APIs.

One approach that has gained increased interest during the recent decade due to the development in nanotechnology is to formulate amorphous APIs in mesoporous materials [6,7,8]. Amorphous substances generally have higher apparent solubility values compared to their crystalline counterparts and can therefore be used in order to obtain a desired therapeutic effect. However, because of their metastable nature, amorphous APIs are driven to recrystallize to the more energetically favorable crystalline form if they are not stabilized in the formulation [9]. When incorporated into a mesoporous structure, the recrystallization of the amorphous API is suppressed due to geometrical constraints, changes in nucleation mechanisms and kinetics inside the small pores, and interaction between the API and the pore walls [10]. Several other approaches have been investigated for physically stabilizing the APIs in its amorphous form. The most commonly used method is to stabilize it through solid dispersion, where the API is mixed with a water-soluble component such as organic polymers (hydroxypropyl methyl cellulose (HPMC), polyethylene glycol (PEG), and polyvinylpyrrolidone (PVP)). The polymer network reduces the mobility of the API, which hinders it from recrystallizing [1,11,12,13]. However, this approach is associated with limited stability of the products, both in the dry state and during dissolution, with recrystallization of the API as a result.

Mesoporous magnesium carbonate (MMC) is an X-ray amorphous material that is synthesized without the use of any surfactants as pore-templating agents. The synthesis route is proven to be scalable, which makes it industrially relevant [14,15]. The average pore diameter of the material can be tuned between 2 and 20 nm [16]. To date, we have demonstrated that this type of MMC material can conserve the amorphous state of a number of different poorly soluble drugs, resulting in enhanced apparent solubility and dissolution rate when tested in vitro [10,16,17]. The increased dissolution rate for celecoxib-loaded MMC was also observed to translate to increased transfer of celecoxib over a caco-2 cell membrane in vitro, mimicking the membrane in the small intestine [18].

The aim of this work was to investigate the in vivo bioavailability of celecoxib formulated with MMC. To the best of our knowledge, this is the first in vivo study of a drug formulation based on MMC used as an excipient to increase the apparent solubility and bioavailability of a drug. Celecoxib is a selective cyclo-oxygenase 2 enzyme inhibitor, widely used for treatment of rheumatoid arthritis, osteoarthritis, and acute pain, and was selected as a model drug for being a poorly soluble and highly permeable compound (e.g., biopharmaceutics classification system II compound). In this study, the MMC formulation was orally administrated in male rats, and the pharmacokinetic properties were studied and compared to Celebra^®^, a commercially available formulation.

## 2. Results

### 2.1. Material Characterization

The MMC material used in this work was synthesized with the same MgO:methanol ratio as in previously published papers [10,19,20,21]. Previous results show that the synthesis of this material is reproducible. Briefly, it has been shown that MMC is X-ray amorphous and that the MMC material consists of particles that are in the microscale. SEM analysis has found that these particles have an irregular structure and that their morphology is not affected by the drug-loading procedure. These results have been discussed in detail previously [10,19,20,21]. Because of the great reproducibility of the MMC synthesis procedure, we did not perform all material characterization techniques in this work but rather focused on N_2_ sorption analysis and DSC in order to confirm the porosity of the material and the crystallinity state of the loaded drug.

The N_2_ adsorption–desorption isotherm in Figure 1a shows that the unloaded MMC sample was highly porous with a BET surface area of 489 m^2^/g and a total pore volume (at p/p_0_ = 0.98) of 0.75 cm^3^/g. The DFT pore size distribution displayed in Figure 1b has a major peak at 5.0 nm, suggesting that the majority of the pores of MMC had a diameter of 5.0 nm. Some micropores with a diameter of 1.4 nm were also detected in the MMC particles. These results are consistent with the pore size analysis of MMC produced in our previous study [16].

Figure 2 displays DSC patterns for unloaded MMC, celecoxib-loaded MMC, and crystalline celecoxib. The endothermic event at 163 °C for the free celecoxib corresponds to the melting point for the crystalline material. The lack of an endothermic event at the same temperature for the loaded MMC sample confirms that the incorporated celecoxib was not present in a crystalline state. The celecoxib loading degree in the MMC formulation was analyzed by RP-HPLC analysis and was determined to be 25 wt%. This was the maximum loading of celecoxib in MMC without giving any endothermic peak at 163 °C. DSC studies showed that celecoxib was mainly amorphous in the loaded MMC sample, since there were no signs of either endo- or exothermic events in the DSC thermograms. These results are in agreement with and supported by Zhang et al. [19].

### 2.2. In Vitro Drug Release Test

The time-dependent release profiles of celecoxib from MMC and the commercially available formulation at pH 1.2, 4.5, and 6.8 are shown in Figure 3. The release of both formulations had similar profiles at both pH 4.5 and 6.8. During the first 10 min, the release of celecoxib from MMC was enhanced 4- and 5-fold at pH 4.5 and 6.8, respectively, compared to the release of celecoxib from the commercially available formulation under the same conditions. After reaching a maximum of 7.9 and 8.5 mg/L dissolved amount of drug after 10 min at pH 4.5 and 6.8, respectively, a decrease in the release profiles was observed. This decrease indicated recrystallization of the dissolved celecoxib in the solution.

The release of celecoxib from the commercially available formulation reached a plateau instantly at around 2 mg/L of dissolved celecoxib and stayed constant during the entire time period.

At pH 1.2, the MMC dissolved due to the acidic condition; hence, the drug load was released both via dissolution of the carrier and diffusion out of the pores. The lack of an initial peak concentration observed at pH 1.2 could possibly be explained by different release kinetics.

### 2.3. In Vivo Absorption of Celecoxib

The bioavailability of celecoxib formulated in MMC and the commercial formulation was determined in male rats. All animals dosed with celecoxib were systemically exposed to the test compound. The mean plasma concentration–time curves are presented in Figure 4, and a summary of the calculated pharmacokinetic parameters is in Table 1. The plasma concentration–time curves for both formulations were similar. The average C_max_ of celecoxib in the commercial formulation was somewhat higher (1160 μg/L vs. 875 μg/L), and the C_max_ occurred on average somewhat earlier (t_max_: 1.67 h vs. 2.33 h) compared to the MMC formulation. The animals administered with the MMC formulation had an 89% relative bioavailability compared to the commercial formulation (relative bioavailability, F_rel_, is a comparison of the AUC between the two different groups), see Table 1. The difference in AUC_inf_ was, however, not statistically significant (*p* > 0.05) between the two formulations (*t*-test). No differences were seen regarding the elimination half-life or mean residence time.

## 3. Discussion

Celecoxib was successfully loaded in MMC to form a formulation containing 25 wt% of the drug. DSC results indicate that celecoxib was preserved in its amorphous state and that no reaction with the carrier was detected, since there were no signs of either endo- or exothermic events in the thermograms.

A higher amount of amorphous celecoxib was released faster in vitro from the MMC carrier compared to the commercially available formulation. After reaching a maximum dissolved concentration of 7.9 and 8.5 mg/L amorphous celecoxib at pH 4.5 and 6.8, respectively, a decrease in the release profiles was observed, indicating recrystallization of celecoxib. The lack of an initial peak concentration observed at pH 1.2 could possibly be due to MMC dissolution in acidic conditions. The results from the in vivo study showed similar drug absorption, where there was no significant difference in the AUC (bioavailability) between the two groups.

These results were obtained with a non-optimized material, where MMC was used as synthesized. This material benefits from a simple synthesis and drug-loading procedure, and the results were achieved without the use of any other formulation additives other than MMC. By tuning the pore and particle size of the material, it would be possible to tailor the release profile of the loaded drug, since previous studies have shown that the drug release is pore and particle size-controlled [16,20].

In conclusion, mesoporous materials with different sizes and pore structures have been evaluated as suitable drug carriers for solubility enhancement. Mesoporous silica materials are one of the most well-studied porous materials for biomedical applications, since they have great ability to enhance the apparent solubility of drugs. However, there are some challenges in connections with the scale-up process of the synthesis of the materials. The mesoporous silica industry struggles with high manufacturing costs due to expensive silica sources and surfactants used in the fabrication. Additionally, the reproducibility of the synthesis at the larger and industrial scale is difficult to control [22,23]. In contrast, the synthesis of MMC is simpler, since there is no need for the use of surfactants, and scalable, making it possible to manufacture MMC in industrial batch sizes.

These results presented herein indicate potential for further development of MMC as a carrier of poorly soluble compounds.

## 4. Materials and Methods

### 4.1. Materials

Magnesium oxide (MgO) was obtained from Sigma-Aldrich. Methanol and ethanol were purchased from VWR International, Spånga, Sweden. Celecoxib was purchased from 3Way Pharm Inc, Shanghai, China. Commercially available celecoxib (Celebra^®^, Pfizer, Brooklyn, NY, USA) was purchased from Apoteket Kronan, Apoteket AB, Solna, Sweden. All chemicals were used as received. Celebra^®^ 100 mg capsules contained lactose monohydrate, sodium lauryl sulfate, povidone, croscarmellose sodium, and magnesium stearate.

### 4.2. Synthesis of MMC

Synthesis of MMC was carried out by dispersing 15 g of MgO in 225 mL of methanol under stirring in a 354 mL Lab-Crest^®^ glass reaction vessel (Andrew Glass Company, Vineland, NJ, USA). When the mixture appeared homogenous, the reaction vessel was sealed, and 4 bar of CO_2_ gas was applied to the vessel. The sealed and pressurized reaction mixture was left stirring for 24 h at room temperature. After 24 h, the CO_2_ pressure was released from the reaction vessel. The reaction mixture was centrifuged at 4696× *g* (5000 rpm) for 60 min to separate the unreacted MgO particles. The MgO-free reaction mixture was dried into a powder under mechanical stirring at room temperature (20–25 °C; 60–100 rpm) in a ventilated fume hood. The obtained powder was dried at 85 °C for 6 h, then at 150 °C for 3 h, and finally at 250 °C for an additional 6 h, all under a flow of N_2_ (20 cm^3^/min).

### 4.3. N_2_ Sorption Analysis

N_2_ sorption was performed on synthesized MMC using an ASAP 2020 surface area analyzer (Micromeritics Instrument Corporation, Norcross, GA, USA). Prior to the sorption experiments, the sample was pre-treated by heating to 100 °C under dynamic vacuum for 6 h to remove any adsorbed water. The N_2_ adsorption and desorption isotherms were recorded at −196 °C for a relative pressure (p/p_0_) range of 0–0.98. The Brunauer–Emmett–Teller (BET) surface area was calculated using the BET equation using the adsorption points at p/p_0_ = 0.05 to 0.15. Pore size distribution was calculated using the Micromeritics Microactive software by adopting the density function theory (DFT, slit pore model) on the adsorption isotherm.

### 4.4. Drug-Loading Procedure

Three grams of celecoxib was dissolved in 150 mL ethanol, after which 10 g of MMC was added and the solvent was evaporated at 75 °C using a rotary evaporator. The synthesized MMC particles were in the micrometer scale, and after the celecoxib loading, the material was sieved with a ≤100 μm sieve in order to obtain samples with a more controlled particle size distribution. The drug-loaded MMC was stored at 70 °C to avoid adsorption of moisture.

### 4.5. Drug-Loading Analysis

The drug-loading analysis was conducted by RISE Research Institutes of Sweden, Södertälje, Sweden and it was conducted as follows: 60 mg of the celecoxib formulation was accurately weighed into 200 mL volumetric flasks. Approximately 75 mL of 0.1% trifluororacetic acid in water was added, and the sample was ultrasonicated for 5 min. A total of 75 mL of acetonitrile was added, and the sample was dissolved by ultrasonication. Diluent (acetonitrile:water, 50:50) was added to the volume, and the drug-loading degree of the MMC formulations was analyzed through RP-HPLC (Acquity UPLC System, Waters, Sollentuna, Sweden) at 250 nm. The HPLC system consisted of a column (Acquity BEH C18 50 × 2.1 mm, 1.7 μm, Waters, Sollentuna, Sweden), and the temperature of the column was 40 °C. Mobile phase A consisted of 0.03% trifluororacetic acid in water, and mobile phase B consisted of 0.03% trifluororacetic acid in acetronitrile. The flow rate of the mobile phase was 0.6 mL/min. Samples were prepared in duplicates.

### 4.6. Differential Scanning Calorimetry

Differential scanning calorimetry (DSC) analyses were performed with a DSC Q2000 instrument (TA Instruments, New Castle, DE, USA) on MMC before and after the incorporation of the drug. In addition, the crystalline celecoxib was also studied by DSC. Samples of 4.9–7.8 mg were weighed into 5 mm aluminum pans and were hermetically sealed. Samples were first cooled to −35 °C for stabilization and then heated to 250 °C at a heating rate of 10 °C/min. Indium (melting point of 156.6 °C and melt enthalpy of 28.4 mJ/mg) was used to calibrate the temperature scale and the enthalpic response.

### 4.7. In Vitro Drug Release Test

The release of both celecoxib formulations was analyzed using a Sotax AT7 Smart USP-2 dissolution bath (Sotax AG, Aesch, Switzerland) equipped with 1000 mL vessels (37 °C, 50 rpm) and paddles as stirring elements (paddle apparatus). Samples with a total content of 100 mg celecoxib were placed in the vessels containing 1000 mL dissolution buffer pH 1.2, pH 4.5 or pH 6.8 (dissolution buffer pH 1.2 TS, pH 4.5 TS2, and pH 6.8 TS were prepared according to the International Pharmacopoeia). Aliquots of 2 mL were withdrawn from each vessel at regular time intervals for 300 min and filtered through 0.45 µm PTFE syringe filters (Merck, Solna, Sweden) prior to analysis. The celecoxib concentration was analyzed using a UV absorbance spectrophotometer at 252 nm. The aliquots were returned to the vessels after each time measurement. The measurements were made in triplicates for both formulations.

### 4.8. In Vivo Drug Absorption of Celecoxib

The in-life phase of the study was performed at Adlego Biomedical AB in Solna, Sweden. Male Sprague Dawley rats were divided into two groups (*n* = 6), where the first group was administered with a single dose of 6 mg/kg commercially available celecoxib (Celebra^®^, Sollentuna, Pfizer) and the second group received a single dose of 6 mg/kg MMC-loaded celecoxib. The weight in kg indicates the body weight of the rats. The commercial formulation consisted of crystalline lactose monohydrate, povidone, croscarmellose sodium, sodium lauryl sulfate, and magnesium stearate.

The doses were shipped to the test site as ready-weighted doses into vials, with one vial per animal. Prior to administration, 1.25 mL of phosphate buffer saline was added to each vial containing the formulations and administered as an oral gavage. Blood samples were collected from the tail vain at regular time intervals (0, 0.25, 0.5, 1, 2, 4, 6, 8, 24, 48, and 72 h) for 72 h after administration, and the celecoxib plasma concentration was analyzed by an LC-MS/MS assay (LLOQ of the assay was 0.1 μg/L for celecoxib in plasma).

The LC-MS/MS assay was performed as follows: plasma samples were transferred to Waters 96-well plates and precipitated with 2-fold volumes of internal standard solution (acetonitrile containing 50 ng/mL of sulfasalazine as an internal standard). The plate was mixed for three min at 600 rpm and centrifuged for 15 min at 4000 rpm. The supernatants were collected into a Waters UPLC 96-well plate for analysis in a Waters Acquity UPLC + Waters TQ-S triple quadrupole MS with a Waters Acquity BEH C18 (2.1 × 50 mm, 1.7 µm) column with precolumn filter.

The standard samples were prepared into blank rat plasma (commercial plasma was used, since about 1.4 ng/mL celecoxib was observed in blank plasma provided by the Sponsor) by spiking into concentrations of 0.2, 0.5, 1, 2, 5, 10, 20, 50, 100, 200, 500, 1000, 2000, 5000, and 10,000 ng/mL (90 µL blank plasma + 10 µL diluted working solutions) of celecoxib. Standards were otherwise treated identically to the samples.

QC samples were prepared by spiking the blank rat plasma into concentrations of 3, 20, 200, and 2000 ng/mL. QC samples were prepared into both commercial blank plasma and customer blank plasma. After spiking, the samples were treated as the standard samples. One replicate QC sample was prepared on each 96-well plate. The collected in vivo data were analyzed by non-compartmental analysis in PhoenixTM WinNonlin^®^ version 6.4, build 6.4.0.768 (Pharsight^®^, St. Louis, MO, USA).

The pharmacokinetic parameters were calculated on full individual profiles, and the analysis consisted of an assessment of pharmacokinetic parameters including maximum measured plasma concentration (C_max_), time to reach maximum plasma concentration (T_max_), terminal half-life (T_1⁄2_), mean residence time extrapolated to infinity (MRT), area under the plasma concentration–time curve (AUC), and the relative bioavailability of the drug (F_rel_). Ethical approval was conducted by regional animal experimental ethics in Stockholm (North), N81/14. Data were analyzed by a *t*-test, and *p*-values lower than 0.05 were considered statistically significant.

## Figures and Tables

**Figure 1 molecules-27-06188-f001:**
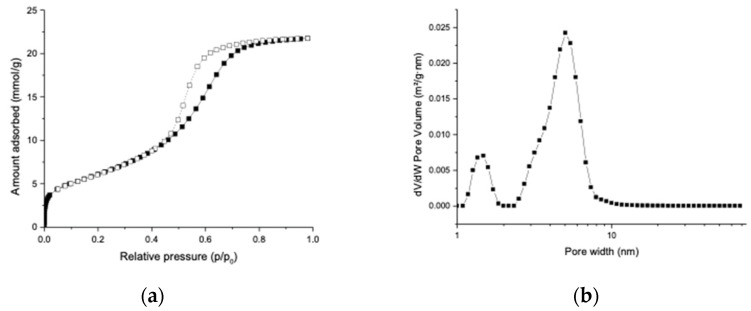
(**a**) N_2_ adsorption isotherm of unloaded MMC recorded at −196 °C. (**b**) DFT pore size distribution of unloaded MMC calculated using the N_2_ adsorption isotherm.

**Figure 2 molecules-27-06188-f002:**
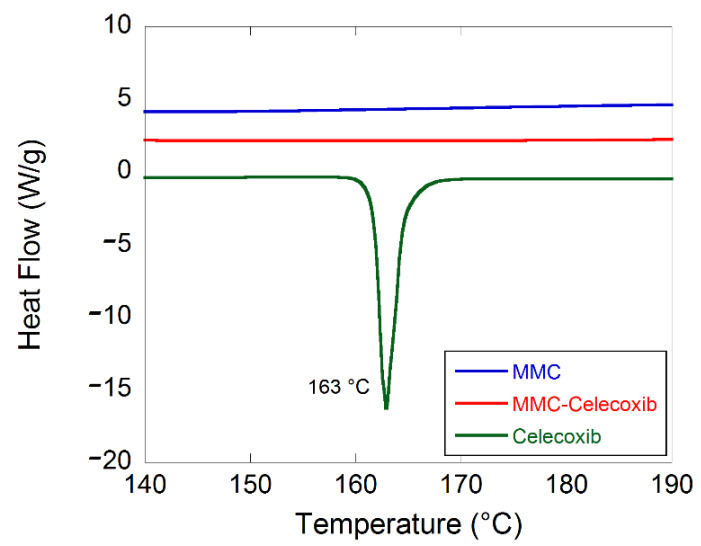
DSC thermograms for the three studied samples.

**Figure 3 molecules-27-06188-f003:**
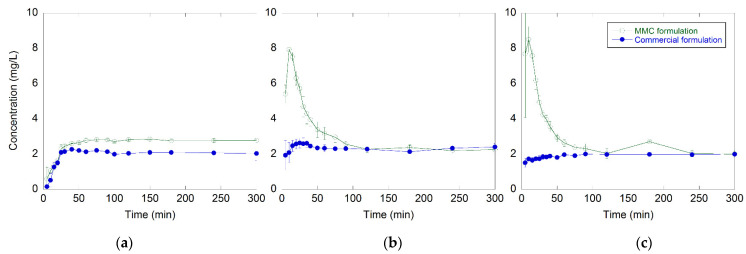
(**a**) Celecoxib dissolution profiles for the MMC and the commercially available formulation at pH 1.2, (**b**) 4.5, and (**c**) pH 6.8. All measurements were made in triplicates, and data are displayed as the mean values with corresponding standard deviations.

**Figure 4 molecules-27-06188-f004:**
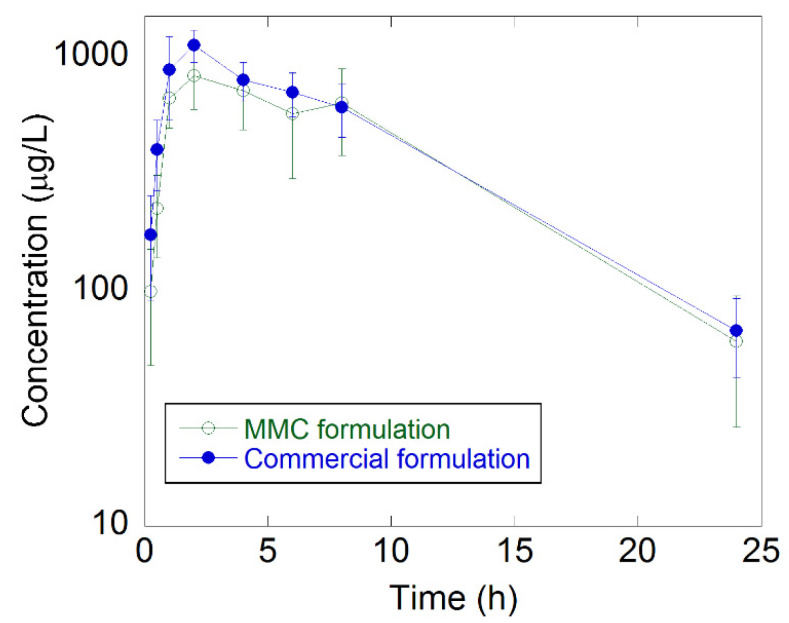
Plasma concentration–time curves for the MMC and the commercially available formulation. The error bars are based on *n* = 6.

**Table 1 molecules-27-06188-t001:** Summary of calculated pharmacokinetic parameters. The values are an average of *n = 6* animals per group. Abbreviations: C_max_, maximum concentration; T_max_, time to reach maximum concentration; T_1/2_, terminal half-time; AUC, area under the curve.

Group	Dose (mg/kg)	C_max_ (μg/L)	T_max_ (h)	T_1/2_ (h)	AUC (h μg L^−1^)	F_rel_ (%)
Commercial formulation	6	1160 ± 182	1.67 ± 0516	6.4 ± 1.19	10700 ± 2160	-
MMC formulation	6	875 ± 220	2.33 ± 0.816	6.12 ± 1.37	9480 ± 3470	89

## Data Availability

The data used to support the findings of the present study are available from the corresponding author (T.Z.G.) upon request.

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
