# Peer review of "Bioavailability of Celecoxib Formulated with Mesoporous Magnesium Carbonate—An In Vivo Evaluation"

_molecules, 2022, doi:10.3390/molecules27196188_

Round 1

Reviewer 1 Report

Dear authors,

I would like to congratulate you on the well written communication. Please find just minor editing suggestions and curiosity questions below.

line 75: Would it be possible to rephrase this sentence for better readability?

line 76: Could you please correct "shows" to "show"

line 239: Was the real pH 1.4.5?

To which glass forming ability class (GFA) does celecoxib fall?

Do you expect a different outcome with a poor glass forming drug?

Would a drug loading of 25% be sufficient for a potential solid oral dosage form?

Did the authors consider using precipitation inhibitors to prevent the drug from crushing out in solution

Kind regards

Author Response

We hereby re-submit our manuscript entitled Bioavailability of celecoxib formulated with mesoporous magnesium carbonate – an in vivo evaluation(Manuscript ID: molecules-1904409) by Teresa Zardán Gómez de la Torre, Tuulikki Lindmark, Ocean Cheung, Christel Bergström, and Maria Strømme for publication as an article in Molecules.

We thank the reviewer for the valuable comments. The issues raised by the reviewer have been delt with and included in the revised manuscript where appropriate.

Below you find all questions listed together with our response to each and one of them. Text changes are marked up using the “Track Changes” function in MS Word throughout the revised manuscript.

 Point 1: line 75: Would it be possible to rephrase this sentence for better readability?

line 76: Could you please correct "shows" to "show"

line 239: Was the real pH 1.4.5?

Answer 1: We thank the reviewer for the comment. We have made all the suggested changes.

 Point 2: To which glass forming ability class (GFA) does celecoxib fall?

Answer 2: According to Panini et al. Pharmaceutics 2019 Oct; 11(10): 529 celecoxib fall into Class II based on crystallization from melt and Class III based on rapid solvent evaporation.

Point 3: Do you expect a different outcome with a poor glass forming drug?

Answer 3: No, we do not expect a different outcome. According to Ditzinger et al. Pharmaceutics 2019, 11, 577, poor glass formers are prone to recrystallization in amorphous formulations. One strategy to tackle this instability is to combine the drug with a polymer in an amorphous solid dispersion.

Point 4: Would a drug loading of 25% be sufficient for a potential solid oral dosage form?

Answer 4: It is dependent on the therapeutic dose of the active substance. In this particular case, 25wt% was sufficient.

Point 5: Did the authors consider using precipitation inhibitors to prevent the drug from crushing out in solution

Answer 5: The aim of the present work was to analyze and investigate the outcome of the plasma concentration-times curve for the MMC formulation without any additives as a proof-of-concept. As a next step, it would be interesting and also very relevant to formulate the MMC together with precipitation inhibitors in order to prevent the drug to re-crystallize.

We hope that our manuscript will be consider for further publication in Molecules after our revision.

With sincere regards,

Maria Strømme

Professor of Nanotechnology

Reviewer 2 Report

Line 116. explain the recrystallization phenomenon at the pH, which was observed

Section 2.3. Pharmacokinetics studies were performed for drug release profile, mention the method either it is compartmental or non-compartmental 

Discussion. Why the pH was used as 4.5 & 6.8, because for drug delivery in different dosage form, pH has a vital role, like gastric, enteric etc. explain?

For the comparison purpose, Celebra was used for comparison, the excipients details for this product is not mentioned 

Line 25. The celecoxib loading degree in the MMC formulation was determined to 25.2 wt%. move this sentence to results 

Section 4.7. Provide the USP reference 

Section 4.8. Male Sprague Dawley rats, describe the sex of rats

Line 248. 6 mg/kg commercially...mg/kg body weight?, mention this 

Line 252-254, the sentence are repeating as dose is already mentioned 

LC-MS/MS, details are missing, either were same as used for drug loading analysis, explain it  

Author Response

We hereby re-submit our manuscript entitled Bioavailability of celecoxib formulated with mesoporous magnesium carbonate – an in vivo evaluation(Manuscript ID: molecules-1904409) by Teresa Zardán Gómez de la Torre, Tuulikki Lindmark, Ocean Cheung, Christel Bergström, and Maria Strømme for publication as an article in Molecules.

We thank the reviewer for the valuable comments. The issues raised by the reviewer have been delt with and included in the revised manuscript where appropriate.

Below you find all questions listed together with our response to each and one of them. Text changes are marked up using the “Track Changes” function in MS Word throughout the revised manuscript.

Point 1: Line 116. explain the recrystallization phenomenon at the pH, which was observed

Answer 1: The recrystallization of celecoxib is due to its poor aqueous solubility at the given pH. Due to its metastable nature, the amorphous drug is driven to recrystallize to the more energetically favorable crystalline form if it is not stabilized in the formulation. This is described in the Introduction part.

Point 2: Section 2.3. Pharmacokinetics studies were performed for drug release profile, mention the method either it is compartmental or non-compartmental 

Answer 2: We thank the reviewer for the comment. The method was performed by a non-compartmental analysis. This information has been added to Section 4.8.

Point 3: Discussion. Why the pH was used as 4.5 & 6.8, because for drug delivery in different dosage form, pH has a vital role, like gastric, enteric etc. explain?

Answer 3: We agree with the reviewer that the pH has a vital role. We have therefor included release profiles at pH 1.2 to cover a broader pH range.

Point 4: For the comparison purpose, Celebra was used for comparison, the excipients details for this product is not mentioned 

Answer 4: The reviewer is right and we have now included the details of the excipients in section 4.2

Point 5: Line 25. The celecoxib loading degree in the MMC formulation was determined to 25.2 wt%. move this sentence to results 

Answer 5: The loading degree is stated in the results (Line 104) but we have removed it from the methods.

Point 6: Section 4.7. Provide the USP reference 

Answer 6: Thank you for the comment. The reference is added to Section 4.7.

Point 7: Section 4.8. Male Sprague Dawley rats, describe the sex of rats

Answer 7: The gender of the rats is stated in section 4.8. They were male rats.

Point 8: Line 248. 6 mg/kg commercially...mg/kg body weight?, mention this 

Answer 8: The reviewer is completely right, the weight (kg) indicates the body weight. We have clarified it in the text.

Point 9: Line 252-254, the sentence are repeating as dose is already mentioned 

Answer 9: We have removed the repeating sentence.

Point 10: LC-MS/MS, details are missing, either were same as used for drug loading analysis, explain it  

Answer 10: Thank you for the comment. The procedure for the LC-MS/MS assay is described in section 4.8 (Line 270-276)

We hope that our manuscript will be consider for further publication in Molecules after our revision.

With sincere regards,

Maria Strømme

Professor of Nanotechnology

Round 2

Reviewer 2 Report

The content of the paper was well organized, all the suggested points are incorporated, and easy for the reader to follow the subject discussed, thus support for its acceptance.